# Analysis on the Temperature Field and the Ampacity of XLPE Submarine HV Cable Based on Electro-Thermal-Flow Multiphysics Coupling Simulation

**DOI:** 10.3390/polym12040952

**Published:** 2020-04-20

**Authors:** Yiyi Zhang, Xiaoming Chen, Heng Zhang, Jiefeng Liu, Chaohai Zhang, Jian Jiao

**Affiliations:** 1College of Electrical Engineering, Guangxi University, Nanning 530004, Guangxi, China; yiyizhang@gxu.edu.cn (Y.Z.); xiaomingchen19960@163.com (X.C.); zdsizyy@126.com (C.Z.); 2Electric Power Research Institute, Guangxi Power Grid Corporation, Nanning 530002, Guangxi, China; 15235030639@163.com

**Keywords:** XLPE submarine cable, temperature field, ampacity, seawater velocity, multiphysics coupling

## Abstract

The operating temperature and the ampacity are important parameters to reflect the operating state of cross-linked polyethylene (XLPE) submarine high voltage (HV) cables, and it is of great significance to study the electrothermal coupling law of submarine cable under the seawater flow field. In this study, according to the actual laying conditions of the submarine cable, a multi-physical coupling model of submarine cable is established based on the electromagnetic field, heat transfer field, and fluid field by using the COMSOL finite element simulation software. This model can help to analyze how the temperature and ampacity of the submarine cable are affected by different laying methods, seawater velocity, seawater temperature, laying depth, and soil thermal conductivity. The experimental results show that the pipe laying method can lead to the highest cable conductor temperature, even exceeding the maximum heat-resistant operating temperature of the insulation, and the corresponding ampacity is minimum, so heat dissipation is required. Besides, the conductor temperature and the submarine cable ampacity have a linear relationship with the seawater temperature, and small seawater velocity can significantly improve the submarine cable ampacity. Temperature correction coefficients and ampacity correction coefficients for steady-state seawater are proposed. Furthermore, the laying depth and soil thermal conductivity have great impact on the temperature field and the ampacity of submarine cable, so measures (e.g., artificial backfilling) in areas with low thermal conductivity are needed to improve the submarine cable ampacity.

## 1. Introduction

In recent years, with the change of energy use and the development of the power grid, the submarine cross-linked polyethylene (XLPE) power cable has been increasingly used [1,2]. The submarine cable is the hub connected to the land-based power grid, and the research on the submarine cable transmission technology is conductive to better engineering applications. The accurate calculation of the temperature field and the ampacity of submarine cable is vital for optimizing the cost of submarine cable construction and improving the efficiency of submarine cable utilization [3]. The submarine cable ampacity refers to the current-carrying capacity under steady-state operation at the maximum operating temperature, with a given laying method and environmental conditions. Presently, XLPE is widely used in submarine cables due to its good electrical and thermo-mechanical behaviors, and relatively low cost [4,5]. The maximum heat-resistant operating temperature of XLPE insulation is 90 °C [6]. If the operating temperature of the cable conductor is higher than 90 °C for a long time, the cable will fail [7,8]. In addition, high temperature will accelerate XLPE insulation aging and shorten the cable service life [9,10]. On the contrary, if the operating temperature of the cable conductor is much lower than 90 °C the operating efficiency of the cable will be reduced [11]. Therefore, the conductor temperature will directly affect the aging of the XLPE insulation and the operation of cable.

There are two common methods for calculating the temperature and the ampacity of submarine cable. One is the equivalent thermal resistance method that is based on the international standard IEC 60287 [12,13]. This method has high efficiency and accuracy in calculating the temperature and the ampacity of the direct buried cables. It can satisfy the calculation of the load in the simple scenario, but it is not suitable for the forced scenario [14]. For instance, the IEC calculation cannot address the physical problems coupling with air convection, radiation, and heat transfer, and it will cause large errors in calculation of the multi-loop and complex environment. Another method is the numerical method, including the boundary element method [15], difference method, and finite element method [16,17,18]. The finite element method that is based on COMSOL can simulate actual working conditions and perform the coupling calculation of multiple physical fields. Therefore, it has become the main research method for analyzing the temperature and ampacity of cable in recent years [19].

There are three laying methods of the submarine cable, namely, pipe laying, direct burial laying, and sub-sea laying [20]. For the pipe laying method, the buried submarine cable is put into the PVC pipe, which is generally applied in the landing section of cable. For the direct burial laying method, the submarine cable is directly put in seabed soil, which is generally applied in the shallow sea. For the sub-sea laying, the submarine cable is put in seawater, which is generally applied in the deep sea. Different laying methods have different effects on the temperature and the ampacity of submarine cables. Certain submarine cables with the excessively high operating temperature need heat dissipation in order to prevent waste that is caused by low ampacity. The operation of the submarine cable is closely related to the marine environment; the heat is generated by the loss of conductors, and the solid heat transfer occurs in the cable body and soil when the fluid transfer occurs in water. Changes in seawater velocity and external temperature will change the operating temperature of the cables and affect the cable ampacity. Therefore, it is of great significance to consider the law of electric coupling of the submarine cable in the flow field.

The electro-thermal-flow multiphysics coupling of the submarine high voltage (HV) cable is a complex physical process and it has been a research hotspot. However, most studies only focus on underground cables and land cables from the aspects of electro-thermal, electromagnetic-thermal coupling, etc. Xu et al. used COMSOL software to simulate the influence of various factors on the temperature field of the tunnel cable and pointed out that the laying position in the tunnel is critical to the temperature and current capacity of the cable [21]. Hwang et al. focused on the thermal analysis on underground cable in PVC pipe, and showed that the cable temperature of direct burial laying is considerably lower than that of pipe laying [22]. Based on the research that analyzed the boundary problem in the coupling of multiple physical fields with the electromagnetic-thermal model of underground three-phase power cables, calculating the cable temperature with the finite element method can cover the shortage of IEC standard [17]. Xiong et al. studied the temperature field and air velocity field of cable trench with an irregular distribution, and results showed that the cable temperature of irregular laying is higher than that of normal laying [23]. Zhu et al. used a variety of methods, including the finite element method, in order to analyze the cable ampacity of different laying methods, and put forward the strategy of cable ampacity from the view of insulation itself [24]. The finite element method is used in the research [25] to study the heat transfer characteristics of submarine cables, which showed that the submarine soil temperature has a certain impact on the temperature of submarine cables. Emeana et al. pointed out the heating capacity of cables was related to the physical properties and thermal properties of the sediment while focusing on the effect of sediment on the submarine cable temperature [2]. Hughes et al. proved that the permeability has a significant influence on submarine cable temperature, according to the finite element simulation [26].

In summary, most scholars investigated cables from the perspectives of the electro-thermal and electromagnetic-thermal coupling, with certain environmental parameters and boundary conditions, but the impact of seawater velocity was not fully considered. Most of the studies focused on the analysis of underground cables and land cables. In this study, the research object is the three-core 220 kV AC submarine XLPE cable used in engineering. Combining the fluid field to the coupling of electromagnetic field and heat transfer field, a coupling model that is based on electromagnetic field, fluid field, and heat transfer field is built. The cable ampacity is obtained by iterative calculation using the Newton–Raphson method. The influence of different laying methods, seawater temperature, seawater velocity, laying depth, and soil thermal conductivity on the temperature distribution and the ampacity of submarine cables is investigated.

## 2. Control Equations

In the electro-flow-thermal coupling model, the electromagnetic calculation of the submarine cable uses the magnetic field control equation. The flow of seawater is obtained by the stratospheric flow control equation. The fluid heat transfer control equation and solid heat transfer control equation are used for the finite element calculation of temperature distribution, and the heat source in the heat transfer control equation comes from the Joule heat that is produced by electromagnetic coupling heating.

(1) Applying the electromagnetic physics to the submarine cable, the control equation is:(1)$$\left\{ \begin{array}{l} \nabla \times H=J\\ B=\nabla \times A\\ J=\sigma E+j\omega D+\sigma v\times B+J_{e}\\ E=-j\omega A \end{array}\right.$$
where, H is magnetic field strength, A/m; J is a current density vector, $A/m^{3}$; B is magnetic flux density, T; A is extra-surface magnetic potential, Vs/m; σ is conductivity, S/m; E is electric field strength, V/m; $J_{e}$ is external injection current density, $A/m^{3}$; and, v is speed (Lorenz), m/s. In this equation, the extra-surface magnetic potential *A* is used as a dependent variable to solve the Maxwell–Ampere law.

(2) Applying the stratospheric physics to seawater, the control equation is:(2)$$\left\{ \begin{array}{l} \rho (u\cdot \nabla )u=\nabla \cdot [-\rho I+\mu (\nabla u+{\left(\nabla u\right)}^{T})]+F\\ \rho \nabla (u)=0 \end{array}\right.$$
where, ρ is the density of the fluid, $kg/m^{3}$; u is an velocity vector of fluid, m/s; I is an unit matrix; μ is power viscosity, Pa⋅s; T is the temperature of fluid material, K; and, F is volume force (general default is 0).

(3) Applying the solid heat transfer physics to the model other than seawater, the control equation is:(3)$$\left\{ \begin{array}{l} \nabla \cdot q+\rho_{1}C_{\rho 1}\frac{\partial T}{\partial t}=Q\\ q=-k\nabla T \end{array}\right.$$
where, ρ is the density of the material, $kg/m^{3}$; $C_{P}$ is solid constant pressure heat capacity, J/(kg⋅k); q is local heat flux density, $W/m^{2}$; k is thermal conductivity, W/(m⋅k); and, Q is the heat source in solid materials, W/m.

Applying the fluid heat transfer physics to model in seawater, the control equation is:(4)$$\rho C_{\rho }u\cdot \nabla T+\nabla \cdot q+\rho C_{\rho }\frac{\partial T}{\partial t}=Q$$
where, u is speed of fluid, m/s.

(4) In the cable, the heat source in the solid heat transfer field and fluid heat transfer field is the Joule heat that is generated by the load current, so the control equation of the electromagnetic heating module is:(5)$$\left\{ \begin{array}{l} \rho C_{\rho }u\cdot \nabla T=\nabla \cdot (k\nabla T)+Q_{e}\\ Q_{e}=J\cdot E \end{array}\right.$$
where, $Q_{e}$ is the heat source in the material, W/m; and, T is the material temperature, K.

## 3. The Simulation Model

### 3.1. The Simulated Geometric Model

The research object of this paper is the 220 kV AC submarine XLPE cable. Figure 1 shows the specific structure of the cable, and Table 1, below, illustrates specific parameters. In the simulation, the laying environment of the submarine HV cable is divided into two areas, and the upper is the seawater area when the lower is the soil area. For direct burial laying, the submarine cable is placed in the lower soil area. For sub-sea laying, the submarine cable is placed in the upper seawater area. For pipe laying, the submarine cable is placed in the pipe in the soil area. Figure 2 shows the submarine cable and its laying environment.

In order to improve the computation efficiency of the model, the following assumptions can be made:

(1) The conductivity of the core conductor and metal sheath of the cable changes with temperature, and the change is as follows:(6)$$\sigma =\frac{1}{\rho_{20}(1+\alpha_{20}(T-20))}$$
where, $\rho_{20}$ is the resistivity of a core conductor at temperature 20 °C, Ω⋅m; $\alpha_{20}$ is the resistance temperature coefficient of core conductor, 1/K;

(2) The frequency electromagnetic field of the cable can be treated as the steady field, ignoring the effect of displacement current.

(3) The hysteresis effect of ferromagnetic substances is ignored, treating as homogeneous medium.

### 3.2. The Boundary Conditions and Grid Division of Simulation

The simulation model in this paper uses four physics interfaces.

(1) Electromagnetic field boundary conditions. Applying the load current with an angle difference of 120° to the three-phase conductor of the submarine cable. Using the coil interface to load the current, the coil boundary can be expressed, as:(7)$$J_{e}=\frac{NI_{coil}}{A}e_{coil}$$
where, J is a current density vector, $A/m^{3}$; N is number of turns (N = 1); A is extra-surface magnetic potential, Vs/m; and, $e_{coil}$ is the electric field, V/m.

(2) Fluid field boundary conditions. Seawater flows from the left to the right interface of the water, and the rest of interfaces are closed. In this case, the “inlet” is connected in the front interface of the water, i.e., water flows from front to back along the cable. The seawater velocity can be expressed as:(8)$$u=u_{0}\cdot n$$
where, $U_{0}$ is the rate of legal inflow, m/s; n is an unit direction vector of the interface.

The “exit” is connected in the rear interface of the water, i.e., the interface pressure of the rear interface is 0 Pa. The boundary condition formula is:(9)p=0

(3) Solid heat transfer boundary conditions. Except for the seawater area, the solid heat transfer physics is used in all areas, and the soil area boundary is set at the constant temperature of 20 °C.

(4) Fluid heat transfer boundary conditions. The seawater area uses the fluid heat transfer physics, and a constant temperature is set to the seawater area boundary. The convective heat flux is applied to the upper and lower interface of the seawater area, and the cooling effect of seawater is simulated. The convective heat flux can be expressed, as:(10)$$q=h\cdot (T_{ext}-T)$$
where, q is heat flux, J/s; h is the heat transfer coefficient; $T_{ext}$ is the external temperature, K; and, T is the temperature of the fluid, K.

The simulation model uses the tetrahedron cell to mesh. Refined mesh is applied for the smaller geometric dimensions in the model (submarine cable body), and sparse mesh is applied for the other regions with larger geometric dimensions to reduce the calculations. The number of meshes after the final division is 244,189.

### 3.3. The Calculated Model of Cable Ampacity

In practice, in addition to the cable temperature distribution, the maximum load current that cables can transmit (i.e., the load current that flows into the conductor) is also an key factor of cable operation when the cable conductor operates at the maximum allowable temperature (90 °C) [11,27]. The relationship between the conductor temperature and the current can be expressed by a one-dimensional nonlinear equation. In this study, the cable ampacity is calculated by the Newton– Raphson method using the equation below:(11)$$I_{k+1}=I_{k}-\frac{\Delta \theta_{c}(I_{k})}{\Delta \theta_{c}’(I_{k})}$$
where, $\Delta \theta_{k}(I)$ is the maximum allowable temperature of conductors (90 °C), $\Delta \theta_{k}{}^{’}(I)$ is the first-order differential of the conductor temperature, *I_k_* is the current estimated value of the cable ampacity, and $I_{k+1}$ is the next estimated value of the cable ampacity. Since the temperature differential $\Delta \theta_{k}{}^{’}(I)$ for each iteration cannot be exported, the difference quotient of each iteration is used to approximate the differential, then:(12)$$\Delta \theta_{c}’(I)\approx \frac{\Delta \theta_{c}’(I_{k})-\Delta \theta_{c}’(I_{k-1})}{I_{k}-I_{k-1}}$$

Write the upper formula in discrete form:(13)$$I_{k+1}=I_{k}-\frac{\Delta \theta_{c}(I_{k})}{\Delta \theta_{c}(I_{k})-\Delta \theta_{c}(I_{k-1})}(I_{k}-I_{k-1})$$

The initial value of the current iteration $I_{0}$ can be calculated by the IEC 60287 standard. The corresponding conductor temperature $\theta_{c}(I_{k})$ is derived from the simulation model, and the submarine cable ampacity is calculated by the Newton–Raphson iteration. In the iterative calculation, when the calculated temperature difference $\Delta \theta_{c}(I)$ is less than 0.01 °C, the cable conductor temperature is considered to be stable at the maximum allowable temperature, and the submarine cable ampacity is the load of the cable. The corresponding carrier iterative algorithm is programmed in MATLAB based on the aforementioned analysis of finite element calculation method. The submarine cable ampacity under each parameter is solved by COMSOL with MATLAB simulation. The calculation flow chart of the ampacity (Figure 3) is as follows:

## 4. Results and Discussion

### 4.1. Effects of Laying Method

Laying the three-core submarine HV cable by three methods, namely, direct burial laying, pipe laying, and sub-sea laying. It is assumed that the load current of the submarine cable is 926.3 A, seawater temperature is 20 °C, seawater velocity is 1 m/s, the soil thermal conductivity is 1.05 W/(m⋅k), and the laying depth of submarine cable in the pipe and soil is 200 cm. The effects of different laying methods on the submarine cable temperature and the submarine cable ampacity are studied under these conditions.

Figure 4 shows the temperature distribution of the submarine cable for three laying methods. The conductor temperature (the maximum temperature of the submarine cable) for direct burial laying, pipe laying and sub-sea laying is 69.95, 90.55 and 35.83 °C, respectively.

Figure 5 shows the radial temperature changes of the submarine cable for three laying methods. The overall temperature along the submarine cable radius is the highest by pipe laying, and that is the lowest by sub-sea laying. In the multi-layer structure of the submarine cable, the temperature distribution is uniform. Copper conductor and metal armor have relatively large thermal coefficients, and the temperature in those places is the highest. The insulation layer is the thickest, with the lowest thermal conductivity, and its temperature decreases the most. The shield layer is thin and it closely relies on the conductor and insulation, and its temperature is approximately equal to the temperature of the nearby layer. The internal and external sheath layers are thick, with small thermal coefficients, and its temperature decay is fast. The armored layer is made of steel wire, which is thin and has a large thermal conductivity, and its temperature is almost unchanged. The temperature of the submarine cable can decrease to approximately 10 °C from the inside to the outside for all methods. 

The submarine cable ampacity of three laying methods is calculated and shown in Figure 6. The cable ampacity of sub-sea laying is 39.53% higher than that of direct burial laying, and 47.65% higher than that of pipe laying.

Different laying methods will affect the temperature distribution and the ampacity of the submarine cable. Certain heat dissipation measures should be taken in pipe laying—if the operating temperature of the submarine cable has exceeded the normal operating temperature of the XLPE cable and the cable ampacity is lower than the load current, then the submarine cable will be out of normal operation. The landing section of the submarine cable by pipe laying has weak heat dissipation, and it is the clamping point of the submarine cable transmission line. On the contrary, the cable ampacity of sub-sea laying is the largest, because the cooling effect of seawater is significant and the conductor temperature is the smallest.

### 4.2. Effects of Seawater Temperature

It is assumed that the load current of the submarine cable is 926.3 A, the seawater velocity is 1 m/s, the soil thermal conductivity is 1.05 W/(m⋅k), and the laying depth of submarine cable in the pipe and soil is 200 cm. The seawater temperature is generally around 12–21 °C [28]. This model takes a value every 3 °C from the seawater temperature range (3–30 °C) to study the effects of different seawater temperatures on the temperature and the ampacity of the submarine cable.

Figure 7 shows the change curve of the conductor temperature and the ampacity with different seawater temperature (3–30 °C) for three laying methods. It can be seen that the conductor temperature has a positive linear change when the corresponding ampacity has a negative linear change, with the increase of the seawater temperature. The conductor temperature for sub-sea lying, direct burial laying and pipe laying is assumed as $y_{1}(x)$, $y_{3}(x)$, and $y_{5}(x)$, respectively; and, the cable ampacity for sub-sea lying, direct burial laying and pipe laying is assumed as $y_{2}(x)$, $y_{4}(x)$, and $y_{6}(x)$, respectively. Thus, linear equations that are related to the seawater temperature can be expressed as:(14)$$y_{1}(x)=1.072x+14.44822$$
(15)$$y_{2}(x)=2009.7222-12.8x$$
(16)$$y_{3}(x)=0.6989x+55.97458$$
(17)$$y_{4}(x)=1149.16222-4.40778x$$
(18)$$y_{5}(x)=0.66654x+77.2158$$
(19)$$y_{6}(x)=991.173-3.39683x$$

The sub-sea laying method that has direct contact with seawater is most affected by seawater temperature, and its increase rate of conductor temperature and decrease rate of ampacity reduction are the largest, as 1.072 and 2009.7222, respectively, according to the linear equations above. In comparison, the temperature increase rate and the ampacity decrease rate of the other two laying methods are much smaller (as 0.66654 and 991.42667, respectively, for pipe laying; as 0.6989 and 1149.16222, respectively, for direct burial laying), because medium blocks direct contact with seawater. These formulas can provide a reference for the effect of seawater temperature on the actual operation of submarine cable.

### 4.3. Effects of Seawater Velocity

It is assumed that the load current of submarine cable is 926.3 A, the laying depth of submarine cable in the pipe and soil is 200 cm, seawater temperature is 20 °C and the soil thermal conductivity is 1.05 W/(m⋅k). The maximum seawater velocity is generally not more than 1 m/s [29]. Thus, values between 0–1 m/s are taken as the seawater velocity in this model to analyze the effect of different seawater velocity on the temperature and the ampacity of submarine cable for three laying methods.

When the seawater velocity is 1 m/s, the contour map of the velocity distribution by sub-sea laying is shown in Figure 8a. The white color area indicates that there is no water flow. The velocity is lower at the left and right ends of the submarine cable, and a certain backflow is formed. The velocity increases to 1.184 m/s at the upper end of the submarine cable, and a velocity vortex is formed. Figure 8b shows the contour map of the seawater velocity distribution for the other two laying methods. In this case, seawater flows normally, and the maximum velocity of seawater is 1.02 m/s.

Table 2 shows the conductor temperature and the submarine cable ampacity under different seawater velocity (0–1 m/s) of three laying methods. Physical flows that are caused by seawater velocity can significantly improve the convection heat transfer on the surface of submarine cable. No matter which laying method is used, with the increase of the seawater velocity, the conductor temperature decreases and the ampacity increases. When compared with the static state, the conductor temperature by sub-sea laying can decrease by 44.31 °C with flowing seawater, and the submarine cable ampacity can increase by 779.4 A. For direct burial laying, the conductor temperature can decrease by 6.57 °C, and the submarine cable ampacity can increase by 52.4 A. For pipe laying, the conductor temperature can decrease by 5.6 °C and the submarine cable ampacity can increase by 26.9 A. The decrease of the conductor temperature and the increase of cable ampacity are both related to the change of seawater velocity. The heat dissipation effect of seawater on cables by sub-sea laying is greater than that by the other two laying methods. The conductor temperature and the submarine cable ampacity are changed significantly in the low-velocity interval; however, they are basically unchanged in the high-velocity interval. Therefore, the effect of velocity in the low-velocity interval is much greater than that in the high-velocity interval.

Table 2 shows that the submarine cable temperature reaches the bottleneck and it tends to be stable when the seawater velocity is larger than 1 m/s. In general, the seawater velocity is not larger than 1 m/s, so a stable state is defined when the conductor temperature and ampacity of the submarine cable do not change with the seawater velocity. In this paper, the ratio of the conductor temperature $T_{s}$ in the steady state (v = 1 m/s) to the conductor temperature $T_{m}$ in the stationary state (v = 0 m/s) is defined as the temperature–velocity influence factor $P_{T}$. The ratio of the ampacity $I_{s}$ in the steady state to the ampacity $I_{m}$ in the stationary state is defined as the ampacity–velocity influence factor $P_{c}$, which is:(20)$$P_{T}=\frac{T_{s}}{T_{m}}$$
(21)$$P_{C}=\frac{I_{s}}{I_{m}}$$

*P_T1_* = 2.422 for sub-sea laying, *P_T2_* = 1.0938 for direct burial laying method, and *P_T3_* = 1.0618 for pipe laying, according to Equation (20). $P_{T}$ is taken as the flow correction coefficient of the submarine cable temperature. According to equation (21), *P_C1_* = 1.7915 for sub-sea laying, *P_C2_* = 1.0519 for direct burial laying, and *P_C3_* = 1.03 for pipe laying. *P_C_* is taken as the flow correction coefficient of the submarine cable ampacity. The velocity correction coefficients can reflect the effects of seawater velocity on submarine cable.

### 4.4. Effects of Burial Depth and Soil Thermal Conductivity

For three-core submarine cables laid directly in soil and in the pipe, it is assumed that the cable load current is 926.3A, seawater temperature is 20 °C, and seawater velocity is 1 m/s. The typical laying depth of submarine cable is 50–150 cm [1], and this model takes a value every 30 mm from the laying depth range of (40–250 cm). Besides, the general soil thermal conductivity is 0.58–1.94 W/(m⋅k) [30], and this model takes a value every 0.1 W/(m⋅k) from the thermal conductivity range [0.6–1.2 W/(m⋅k)]. The effects of the laying depths and the soil thermal conductivity on the temperature and ampacity of the submarine cable by direct burial laying and pipe laying are studied. 

Figure 9 shows changes in the radial temperature of the cable for two methods under different laying depths. Whether it is direct burial laying or pipe laying, the radial temperature of the cable will increase as the laying depth increases (40–250 cm). There is a positive correlation between temperature and laying depth. The cable temperature increase is gradually smaller, and reaches the bottleneck. The temperature change of direct burial laying is larger than that of pipe laying. Figure 10 shows changes in the cable ampacity of two laying methods under different laying depths. Similarly, the cable ampacity is inversely proportional to laying depth. When the laying depth is shallow, the heat dissipation effect of seawater is significant, because seawater has a high convection coefficient and good thermal conductivity. The lower the laying depth, the smaller the cable temperature, and the greater the ampacity. However, if the laying depth is larger than 250 cm, the heat dissipation effect of soil gradually replaces that of seawater. In this case, the temperature and the ampacity of submarine cable are not affected by the laying depth and they remain constant.

The seafloor soil has great influence on the temperature field of submarine cables [31]. Figure 11 shows changes in the radial temperature of the cable for two laying methods with different soil thermal conductivity. The greater the soil thermal conductivity, the smaller the overall temperature of the submarine cable. That is, the soil thermal conductivity affects the heat dissipation effect of soil. If the soil thermal conductivity is small, the conductor temperature of both laying methods will exceed the maximum operating temperature (90 °C) of the XLPE cables. The submarine cable ampacity changes of two laying methods of different soil thermal conductivity are shown in Figure 12. The submarine cable ampacity is positively correlated with the seafloor soil thermal conductivity. With the increase of the seafloor heat conductivity, the submarine cable ampacity increases, and the increase gradually slows down. Before planning and designing transmission lines of the submarine cable, a detailed investigation of the thermal conductivity of the seaside soil is essential for determining the submarine cable ampacity. Thus, in the laying area of submarine cable, it is necessary to collect the upper soil samples that can represent the solid phase matrix material and the particle size distribution to detect the water content index of the upper soil and obtain a reliable upper soil thermal conductivity. At the same time, suitable laying materials (e.g., gravel, stone, and mud) need to be selected in sea areas where artificial backfilling is required, so the influence that is caused by the uneven thermal conductivity on the cable ampacity of transmission lines can be minimized.

### 4.5. Comparison with IEC standards and Actual Data

IEC 60827 standard is generally used for the calculation of the cable ampacity under direct burial laying and pipe laying. Figure 13 shows the cable ampacity calculated by the IEC 60287 and obtained by the simulation. The error between the IEC 60827 calculation and the simulation is 0.67% for direct burial laying, and that is 1.7% for pipe laying, which meets the practical needs (error less than 5%).

For the cables of sub-sea laying, there is no algorithm for verification. In this paper, the measured temperature data obtained by a 220 kV submarine cable monitoring system are compared with the simulated temperature, as shown in Table 3. In the simulation model of submarine cable, the seawater velocity is 1 m/s, the seawater temperature is 20 °C and the soil thermal conductivity is 1.05 W/(m⋅k). In the actual works of submarine cables, the temperature measuring point is 2001 m from the coast, the seawater velocity is 1.1 m/s, the seawater temperature is 20.2 °C and the soil composition is normal sand and mud (the soil thermal conductivity is approximately equal to 1 W/(m⋅k)). Errors of the three-phase (including A-phase, B-phase, and C-phase) data between the simulation of the multi-physics coupling model and monitoring system are 3%, 1.51%, and 0.3%, respectively, which also meet the practical needs.

## 5. Conclusions

In this paper, the 220 kV three-core AC submarine HV cable is the research object, and a multi-physical coupling model is established that is based on the electromagnetic field, fluid field, and heat transfer field. Besides, the temperature field and the submarine cable ampacity are iteratively calculated to study the effects of different laying methods, seawater temperature, seawater velocity, laying depth, and soil thermal conductivity on the temperature and the ampacity of submarine cable. The following conclusions were obtained:

(1) The landing section of the submarine HV cable has the highest operating temperature, the corresponding conductor temperature can reach 90.55 °C, which exceeds the maximum heat-resistant operating temperature of XLPE insulation, and the ampacity is lower than the normal load current. If the submarine cable ampacity is set according to the lowest value of ampacity, the transmission efficiency will be affected. Therefore, the cooling measures for the submarine cable need to be strengthened in order to improve the ampacity, such as using laying materials with high thermal conductivity and performing external ventilation and heat dissipation in the landing section.

(2) The seawater temperature significantly affects the conductor temperature and the submarine cable ampacity. Linear equations of seawater temperature and conductor temperature and linear equations of seawater temperature and submarine cable ampacity are proposed. These equations can provide some reference for the effect of seawater temperature on the actual submarine cable temperature and ampacity.

(3) Flowing seawater can reduce the submarine cable temperature and it has a significant heat dissipation function in low velocity. The temperature flow correction coefficient and the ampacity flow correction coefficient of the submarine cable in the steady-state seawater are proposed in this study, which can quantify the effect of the seawater velocity on the submarine cable in practice.

(4) For the submarine cable by direct burial laying and pipe laying, the laying depth has a great effect on the cable temperature and ampacity. The submarine cable will have better heat dissipation and greater ampacity, if the laying depth is shallower. The cooling effect of the soil will replace that of seawater when the laying depth is larger than 250 cm. The soil thermal conductivity has great influence on the cable temperature and ampacity. In practice, it is necessary to investigate the soil laid on the cable, and artificial backfilling is required in areas with low thermal conductivity in order to increase the submarine cable ampacity.

## Figures and Tables

**Figure 1 polymers-12-00952-f001:**
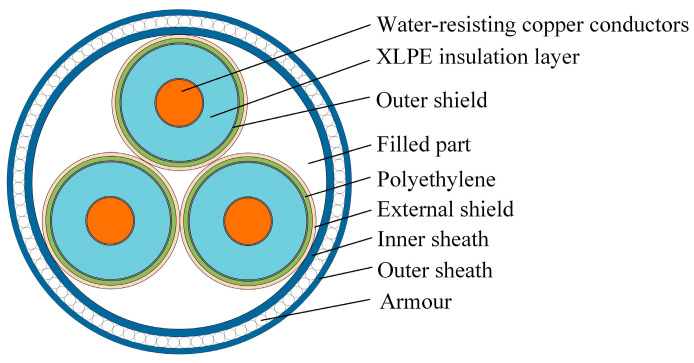
Schematic diagram of submarine cable structure.

**Figure 2 polymers-12-00952-f002:**
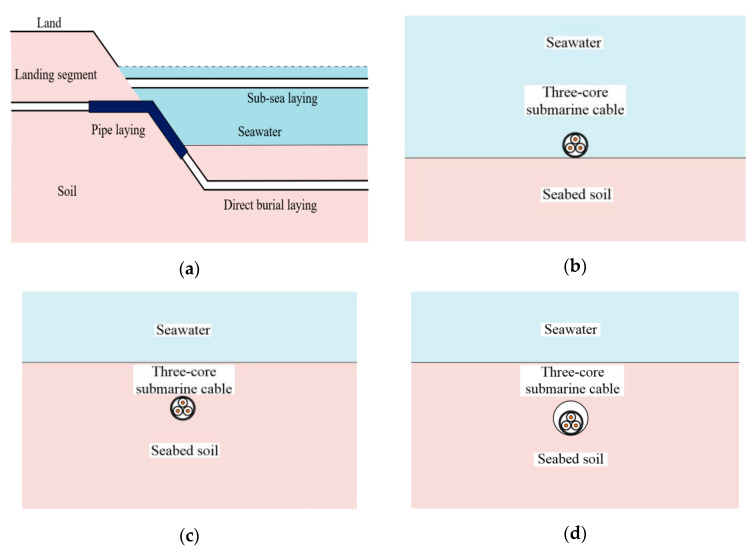
Laying environment of submarine cable. (**a**) Typical laying environment map of submarine cable. (**b**) Sub-sea laying. (**c**) Direct burial laying. (**d**) Pipe laying.

**Figure 3 polymers-12-00952-f003:**
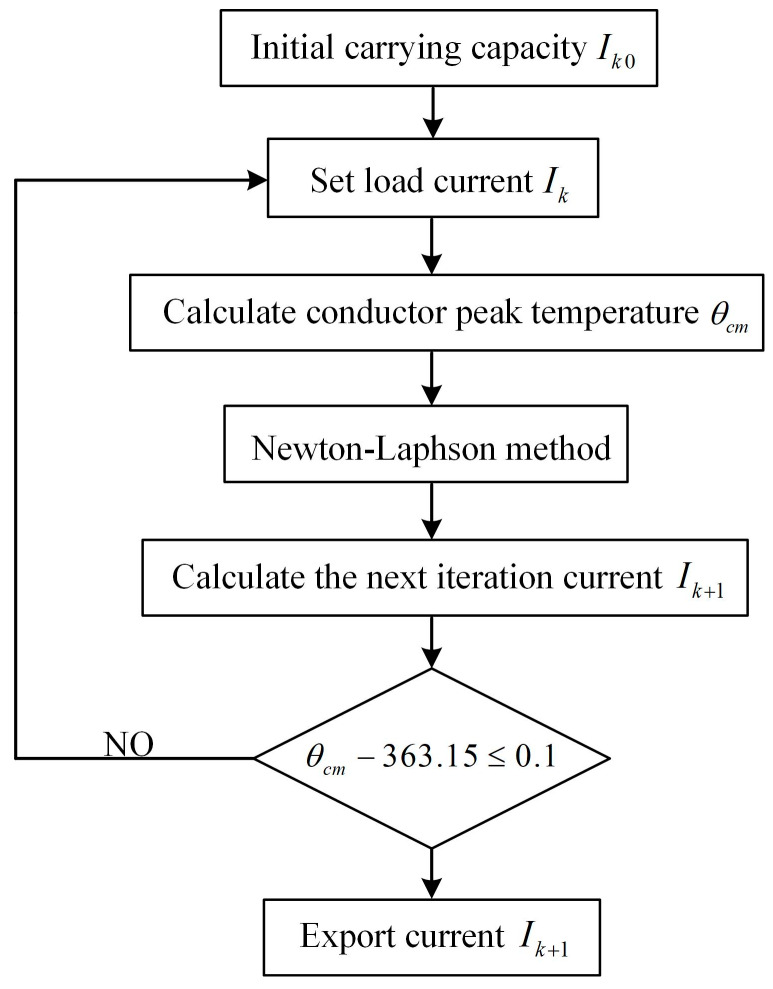
The flow chart of calculated ampacity.

**Figure 4 polymers-12-00952-f004:**
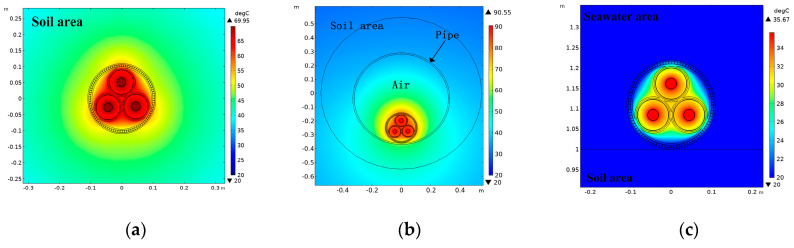
The temperature field distribution of submarine cable for three laying methods. (**a**) The temperature field distribution of submarine cable for direct burial laying. (**b**) The temperature field distribution of submarine cable for pipe laying. (**c**) The temperature distribution of submarine cable for sub-sea laying.

**Figure 5 polymers-12-00952-f005:**
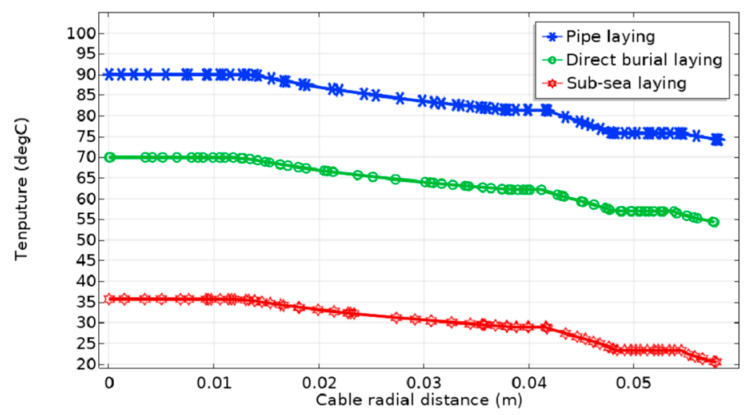
Radial temperature distribution curve of submarine cable for three laying methods.

**Figure 6 polymers-12-00952-f006:**
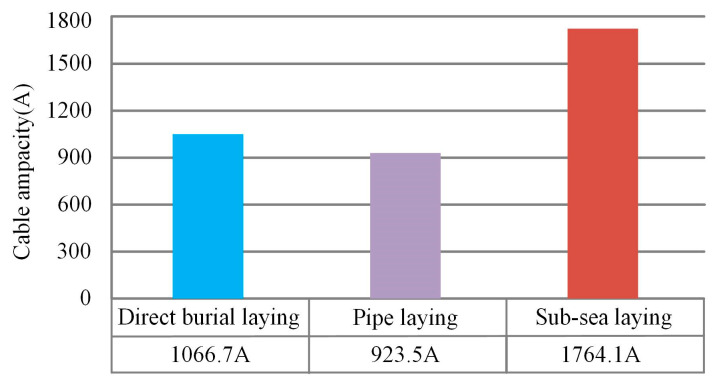
The comparison of the cable ampacity of three laying methods.

**Figure 7 polymers-12-00952-f007:**
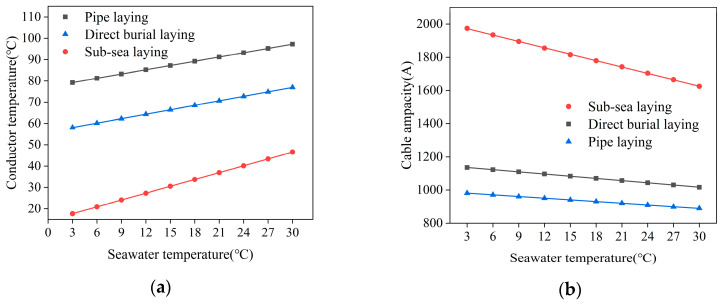
The change curve of the conductor temperature and the submarine cable ampacity. (**a**) The change curve of the conductor temperature for three laying methods. (**b**) The change curve of the submarine cable ampacity for three laying methods.

**Figure 8 polymers-12-00952-f008:**
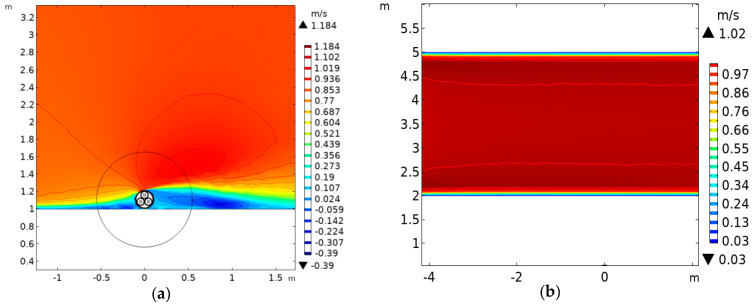
The contour map of seawater velocity distribution. (**a**) The contour map of seawater velocity distribution for sub-sea laying. (**b**) The contour map of seawater velocity distribution for pipe laying and direct burial laying.

**Figure 9 polymers-12-00952-f009:**
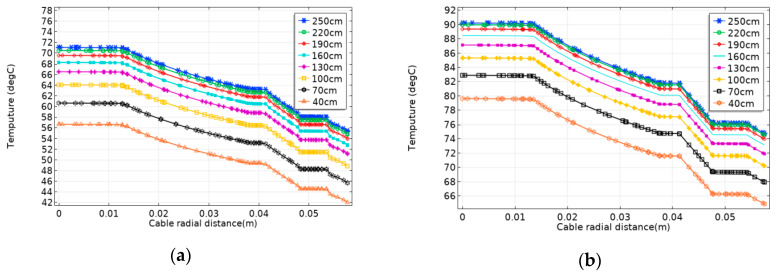
Changes in the radial temperature of the submarine cable for two laying methods with different laying depths. (**a**) Changes in the radial temperature of the submarine cable for direct burial laying with different laying depths. (**b**) Changes in the radial temperature of the submarine cable for pipe laying with different laying depths.

**Figure 10 polymers-12-00952-f010:**
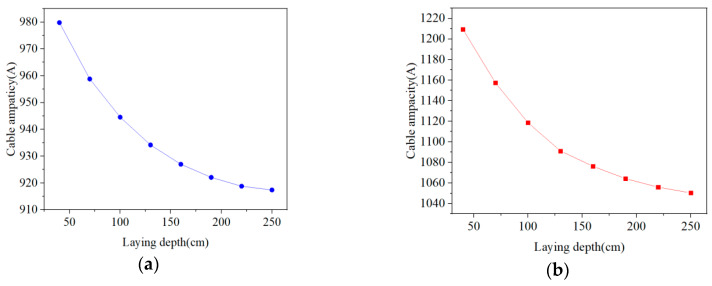
The submarine cable ampacity for two laying methods with different laying depths. (**a**) Submarine cable ampacity for direct burial laying with different laying depths. (**b**) Submarine cable ampacity for pipe laying with different laying depths.

**Figure 11 polymers-12-00952-f011:**
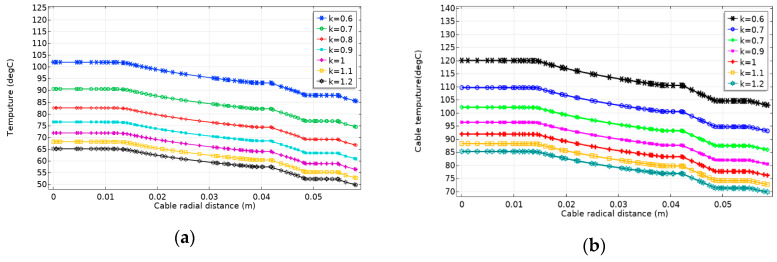
Changes in the radial temperature of the cable for two laying methods with different soil thermal conductivity. (**a**) Changes in the radial temperature of the cable for direct burial laying with different soil thermal conductivity. (**b**) Changes in the radial temperature of the cable for pipe laying with soil different thermal conductivity.

**Figure 12 polymers-12-00952-f012:**
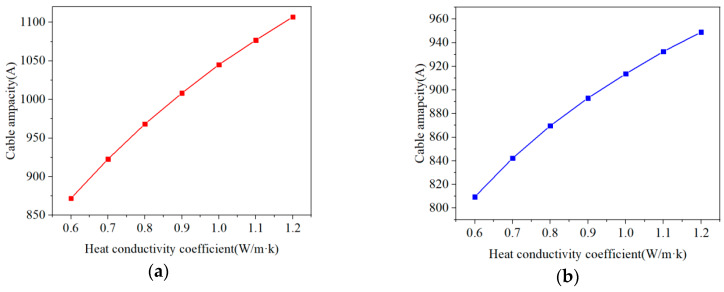
The submarine cable ampacity for two laying methods with different soil thermal conductivity. (**a**) Submarine cable ampacity for direct burial laying with different thermal conductivity of soil. (**b**) Submarine cable ampacity for pipe laying different thermal conductivity of soil.

**Figure 13 polymers-12-00952-f013:**
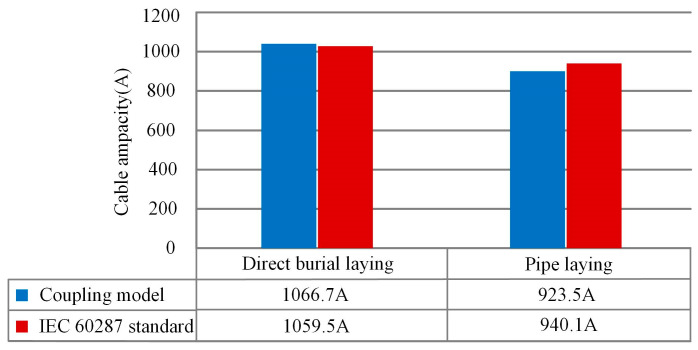
The comparison of the submarine cable ampacity calculated by the IEC 60287 and obtained by the simulation.

**Table 1 polymers-12-00952-t001:** The specific parameters of submarine cable.

Structure	Outer Diameter (mm)	Thermal Conductivity
Conductor	26.6	30.11
Conductor screen	29.7	10
Insulating layer	83.7	2.3
Insulation shielding	86.1	10
Semi-conductive	92.1	10
Lead sheath	99.7	35.3
PE sheath	107.3	0.37
Taped covering	109.3	0.37
Steel wire armor	121.3	44.5
PP rope and brea bed	129.3	0.37

**Table 2 polymers-12-00952-t002:** Conductor temperature and ampacity of submarine cable under different seawater velocity.

Laying Method	Seawater Velocity (m/s)	Conductor Temperature (°C)	Cable Ampacity (A)
Sub-sea laying	0	79.98	984.7
	0.0001	48.70	1352.0
	0.0005	43.63	1469.7
	0.005	39.69	1598.2
	0.010	38.83	1629.9
	0.100	35.78	1759.3
	1.000	35.67	1764.1
Direct burial laying	0	76.52	1008.8
	0.0001	70.27	1058.5
	0.0005	69.97	1061.5
	0.005	69.95	1061.3
	0.010	69.95	1061.2
	0.100	69.95	1061.2
	1.000	69.95	1061.2
Pipe laying	0	96.15	896.6
	0.0001	90.81	922.1
	0.0005	90.58	923.4
	0.005	90.55	923.5
	0.010	90.55	923.5
	0.100	90.55	923.5
	1.000	90.55	923.5

**Table 3 polymers-12-00952-t003:** The measured temperature data and the simulation temperature data.

Data	A-Phase	B-Phase	C-Phase
Monitoring data (**°C**)	34.63	36.21	35.66
Simulation data (**°C**)	35.67	35.66	35.67

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
