# Peer review of "Analysis on the Temperature Field and the Ampacity of XLPE Submarine HV Cable Based on Electro-Thermal-Flow Multiphysics Coupling Simulation"

_polymers, 2020, doi:10.3390/polym12040952_

Round 1
Reviewer 1 Report
The manuscript concerns a very important and current issue related to the influence of the method of submarine cable laying on its temperature field and the ampacity.
Particularly valuable are the results obtained for the cable laid using the sub-sea laying method. In this case the authors showed a significant impact of such parameters as seawater velocity and temperature on temperature field and the ampacity of analysed submarine cable.
In my opinion the results obtained by the authors are valuable and are worth to be published in Polymers.
In my opinion, the article has some shortcomings and needs to be improved before being accepted for publication.
Dear authors, please refer to the following suggestions and comments:
Lines 77-92 – In my opinion, in the literature review you should pointing out the most important achievements of individual authors
Lines 96-98 – Please improve the following sentence: „In this study, add fluid fields based on the coupling of electromagnetic and heat transfer fields, so a coupling model based on electromagnetic, fluid, and heat transfer fields is built for the XLPE submarine HV cable.”
Line 132 – add space between 220 and kV, check the entire article for this.
Line 244: You wrote that: The seawater temperature is generally around 12°C-21°C, but you take into account in your analysis another temperature range. Could you explain your temperature range choice.
Line 377 – In my opinion you should provide more information about the laying conditions of analyzed 220 kV cable, like seawater velocity, soil thermal conductivity, seawater temperature, and other data important for the analysis
Are you aware of statistics regarding the failure frequency of submarine cables depending on how they are laid.
What are the restrictions associated with sub-sea laying method of submarine cable - exposure to damage.
Reviewer 2 Report
The paper is well written and presented.
The topic is interesting and the carried out analysis are accurate.
1) It is not clear one aspect of the analysis. The thermal parameters of the cable are temperature-dependent and their values changes during the simulation. In order to carried out this variation, a transient analysis is necessary, using for example, the conjugate gradient method.
Which is the approach of the authors to this problem?
